# LIV: Language-Image Representations and Rewards for Robotic Control

**Yecheng Jason Ma** [1]   **Vikash Kumar** [2]   **Amy Zhang** [2]   **Osbert Bastani** [1]   **Dinesh Jayaraman** [1]

## Abstract

Motivated by the growing research in natural language-based task interfaces for robotic tasks, we seek good vision-language representations specialized for control. We posit that such representations should: (1) align the two modalities to permit grounding language-based task specifications in visual state-based task rewards, (2) capture sequentiality and task-directed progress in conjunction with cross-modality alignment, and (3) permit extensive pre-training from large generic datasets as well as fine-tuning on small in-domain datasets. We achieve these desiderata through **L**anguage-**I**mage **V**alue learning (LIV), a unified objective for vision-language representation and reward learning from action-free videos with text annotations. We use LIV to pre-train the first control-centric vision-language representation from large human video datasets such as EpicKitchen with no action information. Then, with access to target domain data, the very same objective consistently improves this pre-trained LIV model as well as other pre-existing vision-language representations for language-conditioned control. On two simulated robot domains that evaluate vision-language representations and rewards, LIV pre-trained and fine-tuned models consistently outperform the best prior approaches, establishing the advantages of joint vision-language representation and reward learning within its unified, compact framework.

## 1. Introduction

What are the key machine learning challenges in building a general-purpose robot? Consider a home robot for non-expert end users. Such a robot must acquire common-sense knowledge applicable to generic homes, permitting it to operate from visual observations with some minimal proficiency right off the shelf. Then, it must be able to quickly and robustly adapt to the specifics of the user's home, conditioning its language understanding in the particular visual context of its new habitat. Finally, it must be able to autonomously learn arbitrary new skills specified by its user, most naturally in plain language.

Motivated by such considerations and the recent advances in large vision and language models (Dosovitskiy et al., 2020; Brown et al., 2020; Radford et al., 2021; Alayrac et al., 2022), there has been a surge of research interest in natural language-based interfaces for vision-based robotic control (Lynch & Sermanet, 2020; Stepputtis et al., 2020; Jang et al., 2022; Shridhar et al., 2022a; Brohan et al., 2022). These robot learning approaches largely draw their representations either from existing vision-and-language models (VLMs) (Shridhar et al., 2022a; Liu et al., 2022; Mees et al., 2022b) that were trained for static image-based tasks on static image-text corpuses, or from independently trained vision models and language models in conjunction (Ahn et al., 2022; Lynch et al., 2022).

Neither of these default representation choices is particularly well-suited for sequential decision making in the language-conditioned visuomotor control setting. We identify three key desiderata for control-aware vision-language representations. The first two deal with qualities of the trained representation: (1) It must **align the two modalities to permit grounding language specifications** for effective task representation, and (2) It must **capture task-directed progress grounded in language** to supply intermediate learning signals for autonomous skill acquisition. The last desideratum is concerned with how these control-aware VLMs must be trained. Language grounding is commonly context-dependent, so effective representations must be domain-aware. On the flip side, domain-specific data is typically expensive to collect and therefore scarce in robotics settings, making any domain-specific fine-tuning of large models challenging. Our third criterion for our vision-language representation then is that: (3) It must permit **both extensive domain-generic pretraining as well as domain-specific fine-tuning** on small datasets.

To achieve all three criteria, we propose **L**anguage-**I**mage **V**alue Learning (LIV), a unified objective for joint vision-language representation and reward learning. LIV can flexibly pre-train representations on arbitrary video activity

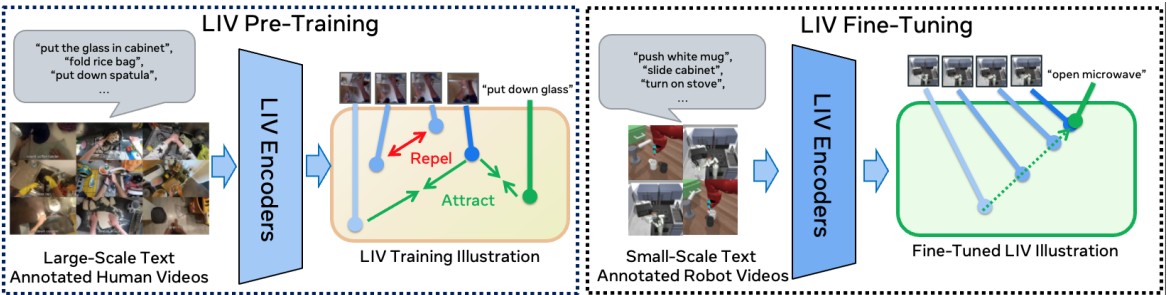

*Figure 1.* **Language-Image Value Learning (LIV)** for vision-language reward and representation learning. Using the same objective for pre-training (Left) and fine-tuning, LIV induces a cross-modal embedding with both temporal coherence and semantic alignment (Right).

datasets with text annotations, even including purely observational datasets of human activity, for which there are several large and conveniently available options (Damen et al., 2018; Grauman et al., 2022; Goyal et al., 2017). Afterwards, the very same objective can be used to fine-tune those representations on small datasets of in-domain robot data, to overcome domain gaps and ground language in context-specific ways.

At a technical level, LIV builds on Value-Implicit Pre-Training (VIP) (Ma et al., 2022b), an approach for learning visual goal-conditioned value functions and representations from videos, generalizing it to learn vision-language values and representations from *language-aligned* videos, as described above. Interestingly, we show that a degenerate instantiation of LIV reduces to the well-known image-text contrastive representation learning objective, as used in CLIP (Radford et al., 2021).

We perform extensive experimental evaluations in two simulated household robotic manipulation settings. Our experiments evaluate LIV vision-language representations not only in their capacity as input state representations for language-conditioned behavior cloning of task policies, but also to directly ground language-based task specifications into visual state-based rewards for robot trajectory optimization, thereby stress-testing alignment across modalities. Along another axis of evaluation, we assess both the "generic" representations pretrained on large human video datasets as well as the specialized representations fine-tuned on in-domain robot data. Our results comparing to several representative recent works from the three distinct categories of pre-training, fine-tuning, and reward learning, confirm the advantages of the LIV objective for joint vision-language representation and reward learning for control.

## 2. Related Work

**Pre-trained Representations for Control.** Our work is related to the literature on pre-training representations for control (Shah & Kumar, 2021; Cui et al., 2022; Parisi et al., 2022; Nair et al., 2022b; Xiao et al., 2022; Ma et al., 2022b;

Fan et al., 2022). These works all seek to use pre-existing data, typically out-of-domain, to pre-train effective representations for downstream unseen robotic tasks. While they all focus on unimodal vision-only representations, Nair et al. (2022b) uses a language alignment loss (Nair et al., 2022a) with respect to a fixed language encoder (Sanh et al., 2019) to shape the visual representation temporally; the learned representation itself is still uni-modal however. In this context, our work is the first to propose a multi-modal vision-language representation pre-training objective for language-conditioned visual control.

On the algorithmic level, LIV builds on value-implicit pre-training (VIP) (Ma et al., 2022b), which casts visual representation and reward learning as a goal-conditioned value function learning problem. LIV extends this approach to the multi-modal vision-language setting and shows a surprising connection to CLIP-style InfoNCE contrastive learning (Oord et al., 2018; Radford et al., 2021).

**Fine-Tuning Pre-Trained Representations.** Several recent works study how to adapt pre-trained representations for downstream tasks (Kumar et al., 2022; Wortsman et al., 2022; Ilharco et al., 2022b; Lee et al., 2022; Kirichenko et al., 2022; Goyal et al., 2022; Dong et al., 2022), motivated by the emergence of large pre-trained models (Radford et al., 2021; Brown et al., 2020) capable of zero-shot transfer. This problem is not resolved even in the standard supervised learning setting with various orthogonal approaches, such as fine-tuning only selective layers (Lee et al., 2022; Kirichenko et al., 2022) and combining several fine-tuned model weights (Wortsman et al., 2022; Ilharco et al., 2022b;a). Most closely related to our work are few concurrent works that find using the CLIP objective to fine-tune CLIP is more effective than alternative fine-tuning approaches (Goyal et al., 2022; Dong et al., 2022). We similarly find LIV objective to be most effective when fine-tuning LIV models and in fact more effective than CLIP fine-tuning for CLIP, demonstrating its full versatility.

**Language-Conditioned Robotic Manipulation.** There has been a surge of interest in language-conditioned vision-based robotic manipulation systems (Lynch & Sermanet,

2020; Stepputtis et al., 2020; Ahn et al., 2022; Jang et al., 2022; Lynch et al., 2022; Shridhar et al., 2022a; Brohan et al., 2022; Shridhar et al., 2022b; Guhur et al., 2022; Liu et al., 2022; Mees et al., 2022b). While several works have considered policy learning on top of pre-trained vision-language representations (Shridhar et al., 2022a; Liu et al., 2022; Mees et al., 2022a), they do not consider how a better representation can be learned in the first place by leveraging large-scale out-of-domain text-annotated video data. As such, (1) how to pre-train new vision-language representations for language-conditioned visuomotor control, and (2) whether doing so is in fact beneficial over existing representations (e.g., CLIP) are open questions that our work first proposes to address.

On the axis of downstream policy learning algorithm, most existing works focus on language-conditioned behavior cloning (LCBC) (Lynch & Sermanet, 2020; Stepputtis et al., 2020). This paradigm demands the expensive collection and text labeling of demonstration data, which can take months to complete (Jang et al., 2022; Lynch et al., 2022; Brohan et al., 2022). In contrast, while LIV is effective as a pre-trained representation for LCBC, it can also be used as a language-conditioned visual reward model that supports autonomous skill acquisition via reinforcement learning (Goyal et al., 2021; Nair et al., 2022a; Mahmoudieh et al., 2022). Our experiments show that LIV outperforms prior state-of-art language-conditioned reward models (Nair et al., 2022a;b) in model-based planning settings.

## 3. Preliminaries & Problem Setting

In this section, we review the VIP algorithm and describe our problem setting.

**Value Implicit Pre-Training (VIP).** VIP (Ma et al., 2022b) learns the optimal goal-conditioned value function via the dual goal-conditioned RL formulation (Ma et al., 2022a;c):

$$\mathcal{L}(\phi) = \mathbb{E}_{p(g)}[(1-\gamma)\mathbb{E}_{\mu_0(o;g)}\left[-\mathcal{S}(\phi(o);\phi(g))\right]$$
$$+ \log \mathbb{E}_{(o,o';g)\sim D}\left[\exp\left(\mathcal{S}(\phi(o);\phi(g)) + 1 - \gamma\mathcal{S}(\phi(o');\phi(g))\right)\right]],$$
$$(1)$$

where $\mu_0(o;g)$ is the distribution of initial frame conditioned on the goal frame $g$ and $D(o,o';g)$ is the goal-conditioned distribution of two successive intermediate frames. In VIP, the value function is implicitly parameterized as a similarity metric (e.g., $L_2$ distance) in the embedding space $V(o;g) := \mathcal{S}(\phi(o);\phi(g))$, making it both a representation learning and a reward learning algorithm. Since it does not depend on actions, VIP can be pre-trained on large-scale human video datasets. The resulting implicit value function serves the dual purposes of (1) reward specification, and (2) visual representation for unseen robot tasks.

**Vision-Language Representation Pre-Training for Control.** We assume access to a dataset of language-annotated videos $D = \{v_i := (o_1^i, ..., g^i; l^i)\}_{i=1}^N$, where each $o \in O := \mathbb{R}^{H \times W \times 3}$ is a raw RGB image, $g^i$ the last frame of the video, and $l^i$ is the textual description of the transpired event in $v_i$. As the video dataset can be out-of-domain with respect to our robot agent (e.g., human videos), we do not assume access to action labels. Datasets of this nature, such as human daily activity videos (Damen et al., 2018; Miech et al., 2019; Grauman et al., 2022), are readily available for research use. A pre-training algorithm $\mathcal{A}$ ingests this training data and outputs vision-language encoders $(\phi, \psi) := \mathcal{A}(D)$, where the vision encoder $\phi : \mathbb{R}^{H \times W \times 3} \to K$ and the language encoder $\psi : L \to K$ each map to the same $K$-dimensional vision-language representation space.

A standard way to learn a vision-language representation is by aligning the modalities via contrastive learning. Specifically, this semantic alignment is achieved by minimizing the InfoNCE objective (Oord et al., 2018):

$$\mathcal{L}_{\text{InfoNCE}}(\phi, \psi) = \mathbb{E}_{p(o,l)}\left[-\log \frac{e^{\mathcal{S}(\phi(o);\psi(l))}}{\mathbb{E}_{D(o')}\left[e^{\mathcal{S}(\phi(o');\psi(l))}\right]}\right],$$
$$(2)$$

where $\mathcal{S}$ is a choice of similarity metric. Intuitively, this objective aims to attract the representations of matching image-text pairs $(o, l)$, while repelling mismatching pairs. Many state-of-art vision-language models (Radford et al., 2021; Jia et al., 2021; Li et al., 2022) train with this InfoNCE objective at scale to deliver strong zero-shot performance on a myriad of vision-language tasks.

**Language-Conditioned Policy Learning using Pre-Trained Representations.** To evaluate vision-language representations $(\phi, \psi)$, we learn policies for robot tasks specified via language commands $l$. Each such task can be formally instantiated as a Markov decision process $\mathcal{M}(\phi, \psi) := (S, A, R(o_t, o_{t+1}; l), T, \gamma, \psi(l))$, where the state space is the space of observation embeddings $S = \phi(O)$, $T$ the transition function, and $\gamma$ the discount factor. The parameters of $(\phi, \psi)$ are frozen during policy learning, and a policy $\pi : \mathbb{R}^K \to A$ must output actions based on the embedded observation and goal, $a \sim \pi([\phi(o), \psi(l)])$.

## 4. LIV: Language-Image Value Learning

### 4.1. Algorithm

We begin by extending the VIP framework to multi-modal goal specifications. This is straightforward given the goal-conditioned nature of Eq. (1), since we can simply replace encoded image goal $\phi(g)$ with encoded text goal $\psi(l)$ and

optimize for a *multi-modal* VIP objective:

$$\mathcal{L}(\phi, \psi) =$$
$$\underbrace{\begin{aligned}&+ \mathbb{E}_{p(g)}[(1-\gamma)\mathbb{E}_{\mu_0(o;g)}[-\mathcal{S}(\phi(o);\phi(g))]\\&+ \log\mathbb{E}_{(o,o';g)\sim D}\left[\exp\left(\mathcal{S}(\phi(o);\phi(g)) + 1 - \gamma\mathcal{S}(\phi(o');\phi(g)))\right]\right]\end{aligned}}_{\text{VIP-I}}$$
$$\underbrace{\begin{aligned}&+ \mathbb{E}_{p(l)}[(1-\gamma)\mathbb{E}_{\mu_0(o;l)}[-\mathcal{S}(\phi(o);\psi(l))]\\&+ \log\mathbb{E}_{(o,o';l)\sim D}\left[\exp\left(\mathcal{S}(\phi(o);\psi(l)) + 1 - \gamma\mathcal{S}(\phi(o');\psi(l)))\right]\right]\end{aligned}}_{\text{VIP-L}}$$
$$\tag{3}$$

As shown, this objective consists of two independent components; VIP-I (Image) ensures the representation to encode a goal-conditioned value function conditioned on image goal, and likewise, VIP-L (Language) for language goal.

At first glance, the LIV objective does not appear to be directly optimizing for semantic alignment between goals in the two modalities, as the respective modality-specific VIP objective is independently optimized. Without alignment, semantically equivalent goals in the respective modality may actually be distant in the representation space. This is undesirable for reward specification, which requires visual grounding of linguistic task descriptions. Intriguingly, in the next section, we show that such semantic alignment is in fact automatically achieved from optimizing Eq. (3).

### 4.2. Theoretical Analysis

Now, we show that by optimizing (3) with a simple augmentation to the training videos, VIP naturally optimizes semantic alignment. Specifically, if we were to consider a *degenerate* distribution of videos, i.e., videos consisting of solely static text-aligned frames $v = ((o, o); l)$, we recover a discounted variant of the InfoNCE objective from VIP-L:

**Proposition 1.** *Let the video distribution consist of solely degenerate videos of repeated frames that align with the text annotation, $D := \{v := ((g, g); l))\}$. Then, the VIP-L objective is equivalent to the InfoNCE objective up to a constant:*

$$\mathcal{L}_{VIP\text{-}L}(\phi, \psi) = \mathbb{E}_{p(g,l)}\left[-\log\frac{e^{(1-\gamma)\mathcal{S}(\phi(g);\psi(l))}}{\mathbb{E}_{D(g')}\left[e^{(1-\gamma)\mathcal{S}(\phi(g');\psi(l))}\right]}\right] + 1,$$
$$\tag{4}$$

*where $p(g, l)$ is the distribution of goal frame and text pair.*

The proof is in Appendix A. This result, though simple to derive, has several important implications. First, note that Eq. (4) is precisely what CLIP (Radford et al., 2021) optimizes (Eq. (2), modulo the discount factor) by contrastively learning similarity between matching image-text pairs. The fact that this objective can be obtained by optimizing VIP-L with a degenerate video distribution suggests that VIP-L is a natural generalization of the InfoNCE objective to the decision making setting, where the data is temporal. In practice, as we will show, this degenerate video distribution can

be trivially obtained by augmenting any existing annotated video in the dataset by repeating the last frame.

This finding also directly suggests a method for *fine-tuning* pre-trained vision-language models for control: use LIV on in-domain labeled videos such as text-annotated robot demonstrations. Several concurrent works (Goyal et al., 2022; Dong et al., 2022) have suggested that fine-tuning a pre-trained model using the same objective (in particular, using CLIP objective to fine-tune CLIP model) can be more effective than fine-tuning using the downstream task objective. When working with CLIP-like pretrained embeddings which are a popular vision-language representation choice, it is then natural to fine-tune them for control with the LIV objective, which is but a natural extension of CLIP that exploits sequential, goal-directed video data.

As we show in our experiments, using the VIP objective to fine-tune pre-trained CLIP models is far more effective than using the CLIP objective. CLIP fine-tuning aligns the last frame in the video to its text description but fails to leverage earlier frames from the same video sequence. In contrast, VIP fine-tuning makes full use of the dataset and naturally balances between semantic alignment and temporal consistency that are both crucial for effective language-conditioned control.

### 4.3. Implementation

Based on the analysis above, we see that, despite initial appearances, the LIV objective of Eq. (3) does in fact naturally induce semantic alignment between visual and language goals. In particular, LIV is implicitly optimizing for a pathway that connects the two modalities via mutual information maximization. Given this pathway that makes goals interchangeable across modalities, in practice, we elect to optimize the VIP objective in just one modality in conjunction with the vision-language InfoNCE objective in Eq. (4):

$$\mathcal{L}_{\text{LIV}}(\phi, \psi) =$$
$$\underbrace{\begin{aligned}&+ \mathbb{E}_{p(g)}[(1-\gamma)\mathbb{E}_{\mu_0(o;g)}[-\mathcal{S}(\phi(o);\phi(g))]\\&+ \log\mathbb{E}_{(o,o';g)\sim D}\left[\exp\left(\mathcal{S}(\phi(o);\phi(g)) + 1 - \gamma\mathcal{S}(\phi(o');\phi(g)))\right]\right]\end{aligned}}_{\text{VIP-I}}$$
$$\underbrace{+ \mathbb{E}_{p(g,l)}\left[-\log\frac{e^{(1-\gamma)\mathcal{S}(\phi(g);\psi(l))}}{\mathbb{E}_{D(g')}\left[e^{(1-\gamma)\mathcal{S}(\phi(g');\psi(l))}\right]}\right]}_{\text{InfoNCE}},$$
$$\tag{5}$$

Pseudocode is presented in Algorithm 1. In each gradient step, a minibatch of video sub-clip consisting of initial, intermediate, and final frames are sampled along with the corresponding text annotations. These samples are used to estimate the VIP-I and InfoNCE losses, which then update the vision-language architecture via back-propagation.

We have shown above that the LIV objective subsumes

---

**Algorithm 1** Language-Image Value Learning (LIV)

1: **Require**: Offline text-annotated (human) videos $D = \{(o_1^i, ..., g^i; l^i)\}_{i=1}^N$, vision-language architecture $(\phi, \psi)$
2: **for** number of training iterations **do**
3:      Sample sub-trajectories $\{o_t^i, ..., o_k^i, o_{k+1}^i, ..., g^i; l^i\}_{i=1}^B \sim D, t \in [1, h_i - 1], t \le k < h_i, \forall i$
4:      $\mathcal{L}_{\text{VIP-I}}(\phi) := \frac{1-\gamma}{B} \sum_{i=1}^B \left[ -\mathcal{S}(\phi(o_t^i); \phi(g^i)) \right] + \log \frac{1}{B} \sum_{i=1}^B \exp \left[ \mathcal{S}(\phi(o_k^i); \phi(g^i)) + 1 - \gamma \mathcal{S}(\phi(o_{k+1}^i); \phi(g^i)) \right]$
5:      $\mathcal{L}_{\text{InfoNCE}}(\phi, \psi) := \frac{1-\gamma}{B} \sum_{i=1}^B \left[ -\log \frac{e^{(1-\gamma)\mathcal{S}(\phi(g^i); \psi(l^i))}}{\frac{1}{B} \sum_{j=1}^B \left[ e^{(1-\gamma)\mathcal{S}(\phi(g^j); \psi(l^i))} \right]} \right]$
6:      Update $(\phi, \psi)$ using SGD: $\phi \leftarrow (\phi, \psi) - \alpha \nabla (\mathcal{L}_{\text{VIP-I}}(\phi) + \mathcal{L}_{\text{InfoNCE}}(\phi, \psi))$
7: **end for**

---

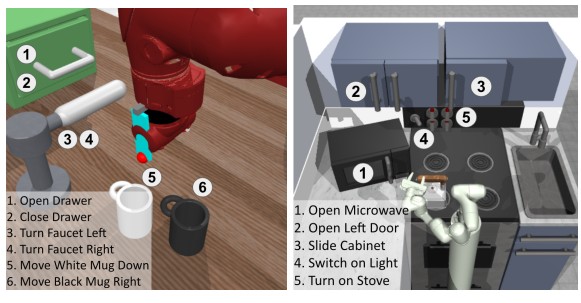

(a) MetaWorld      (b) FrankaKitchen

| Environment | Tasks | Horizon | Dataset Size | Dataset Type |
|---|---|---|---|---|
| MetaWorld | 6 | 20 | 1M | Random |
| FrankaKitchen | 5 | 50 | 12.5K | Demos |

*Figure 2.* Multi-Task Vision-Language Manipulation Benchmarks.

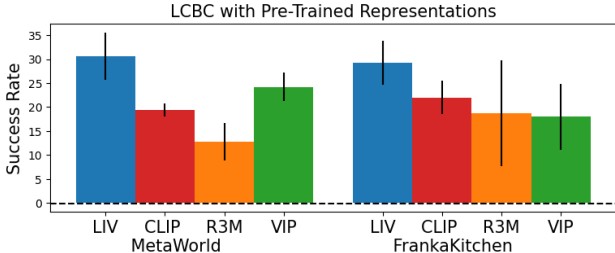

*Figure 3.* **Pre-Trained Representations for Language-Conditioned Imitation Learning:** LIV-EPIC achieves the highest average success rates across two distinct benchmarks.

CLIP-style contrastive objectives. In implementing LIV, we stay close to CLIP architecture and design choices to allow fair comparison to pre-trained CLIP with ResNet50 (He et al., 2016) vision backbone. Finally, we use a $\gamma$-weighted cosine similarity metric for $\mathcal{S}(\phi(\cdot), \psi(\cdot))$ so it represents a valid value function. See Appendix B for details.

## 5. Experiments

Our experiments aim to answer the following questions:

1. Does LIV enable effective vision-language pre-training for control?
2. Can LIV successfully fine-tune pre-existing vision-language models?
3. Can LIV perform language-conditioned reward specification?

We evaluate LIV's effectiveness for pre-training (Section 5.1) and fine-tuning (Section 5.2) by using the resulting representations as the vision-language backbone in language-conditioned imitation learning (LCBC). To assess LIV's reward learning capability, we use its reward function for model-based planning to solve language-specified tasks (Section 5.3).

### 5.1. Pre-Training

**LIV Pre-Training.** We pre-train LIV on EpicK-itchen (Damen et al., 2018), a large-scale dataset of narrated videos of humans completing tasks in diverse household kitchens; we call this pre-trained model **LIV-EPIC**. See Appendix B for details.

**Baselines.** We compare against **CLIP** (Radford et al., 2021), a state-of-art vision-language representation that has seen wide adoption in various robotics tasks (Shridhar et al., 2022a; Cui et al., 2022; Khandelwal et al., 2022; Tam et al., 2022); as LIV is trained using the CLIP architecture, this is the closest comparison. Besides the quantitative LCBC results, we provide qualitative comparison to CLIP in Appendix F to study respective representation's capability of capturing image and language conditioned task progress on unseen EpicKitchen videos, and we find LIV to vastly outperform CLIP in that regard.

We also compare against **R3M** (Nair et al., 2022b) and **VIP** (Ma et al., 2022b), two state-of-art pre-trained visual representations. While unimodal, both are strong baselines; they are pre-trained on ego-centric videos similar to EpicK-itchen (Grauman et al., 2022) and employ the same vision architecture ResNet50 as LIV. We adapt them to the vision-language setting by coupling them with a pre-trained DistilBERT encoder (Sanh et al., 2019) to process language input. We note that R3M does employ this very same model for shaping its visual representation during training, making

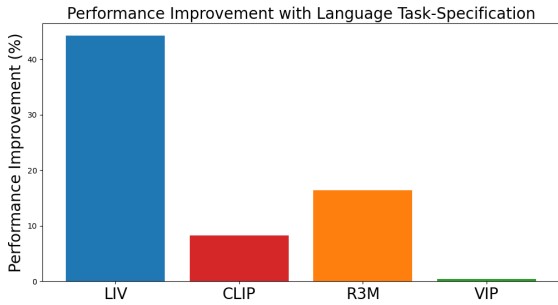

*Figure 4.* **Comparison Between Language vs. One-Hot Task Encoding:** LIV benefits the most from using language task-specification, resulting in near 45% gain in absolute success rates.

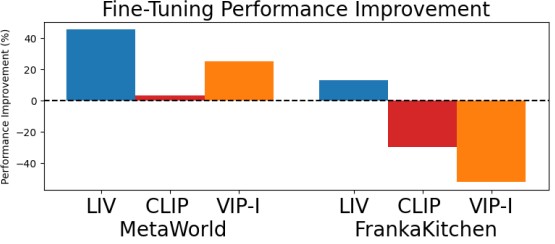

*Figure 5.* **Fine-Tuning Comparisons:** LIV fine-tuning yields significantly higher performance improvement over its ablations.

DistilBERT a natural design choice. To assess the importance of language task-specification and how each method utilizes its language embedding, we also evaluate policy learning using **One-Hot** encoding for tasks.

**Environments.** We consider two multi-task language-conditioned visual manipulation environments, extending the MetaWorld (Yu et al., 2020) and FrankaKitchen (Gupta et al., 2019) benchmarks. The MetaWorld benchmark is taken from Nair et al. (2022a), which has also released a dataset consisting of 1M transitions collected via random actions for policy learning; the trajectories are labeled with task descriptions based on true environment state. The FrankaKitchen benchmark takes existing tasks supported in the environment but makes them specified via fixed task descriptions; we use the tasks and the dataset from Nair et al. (2022b). Each environment has multiple language-specified robot tasks, and includes randomized objects and attributes for each evaluation episode, testing the representations' ability to ground language concepts under visual shifts. The environments and associated tasks are illustrated in Figure 2. See Appendix C for more details on these tasks and datasets.

**Policy Learning and Evaluation.** Following prior works on pre-training for robotic manipulation (Nair et al., 2022b; Ma et al., 2022b; Xiao et al., 2022), we employ a simple MLP architecture on top of the pre-trained representations for the policy network. A single multi-task policy is trained for all tasks within a benchmark. The policy network takes concatenated current observation embedding and language task embedding or one-hot encoding as input, and is trained via behavior cloning on the given benchmark dataset; the representations are kept frozen during policy learning. We evaluate two trained policy checkpoints from when the training is half and fully complete by recording the success rate on 50 rollouts for each task and keep the higher of the two. For each backbone representation, we train policies using 3 random seeds and report the mean and the standard deviation of the success rates. See Appendix D for additional

details on training hyperparameters.

**Results.** Full results are reported in Figure 3 (full numeric results in Appendix D). As shown, our pre-trained LIV-EPIC model, without any in-domain fine-tuning, performs best in both MetaWorld and FrankaKitchen. Furthermore, as shown in Figure 4, LIV-EPIC consistently benefits from its jointly trained language representation. In particular, on both benchmarks, while LIV-EPIC and the strongest baselines (CLIP and VIP) all perform similarly with one-hot encoding, LIV-EPIC realizes much greater gain when language task-specification is used. In fact, using language task-specification *hurts* all baselines on MetaWorld. We hypothesize that this is due to the fact that the MetaWorld dataset contains many episodes whose annotations are long descriptions that consist of concatenation of shorter atomic instructions; for example, `"close drawer turn faucet right push black mug right"` is a valid annotation that contains 3 atononic instructions. Therefore, the language embeddings from pure language model (e.g, DistilBERT) or language model trained from image-text only datasets (e.g., CLIP) may fail to disambiguate these instructions, leading to incorrect task aliasing that hampers policy learning. In contrast, one-hot encoding treats every description as distinct and does not have this aliasing problem. Together, these results highlight the challenges of adapting pure image-text representations and uni-modal visual representations to language-conditioned robotic control, thereby affirming LIV's unique effectiveness in vision-language pre-training for language-conditioned visuomotor control.

## 5.2. Fine-Tuning

Next, we show that the LIV objective can also be used to effectively fine-tune pre-trained vision-language models for downstream policy learning. In particular, we show that it is effective in adapting both pre-trained LIV-EPIC and CLIP models from Section 5.1, despite their vastly different training data and objectives. Specifically, we first take the same in-domain task data as in Section 5.1 to fine-tune the pre-trained representations using the LIV objective (Algorithm 1). Then, as before, we freeze the fine-tuned

*Table 1.* **Fine-Tuning Vision-Language Representations:** LIV consistently improves the performance of pre-trained vision-language models regardless of their initial capabilities, the sizes and the qualities of the in-domain fine-tuning datasets.

**MetaWorld**

| Model/Method | Pre-Trained | **LIV** | CLIP | VIP-I |
|---|---|---|---|---|
| Random | $20.6 \pm_{1.0}$ | $27.8 \pm_{4.1}$ | $30.8 \pm_{2.2}$ | $30.6 \pm_{3.5}$ |
| CLIP | $19.4 \pm_{1.3}$ | $33.9 \pm_{7.5}$ | $16.4 \pm_{4.3}$ | $30.0 \pm_{2.2}$ |
| LIV-EPIC | $30.6 \pm_{5.0}$ | $\mathbf{35.8} \pm_{1.4}$ | $21.4 \pm_{5.7}$ | $20.3 \pm_{3.4}$ |

**FrankaKitchen**

| Model/Method | Pre-Trained | **LIV** | CLIP | VIP-I |
|---|---|---|---|---|
| Random | $17.7 \pm_{3.9}$ | $19.2 \pm_{3.8}$ | $17.1 \pm_{2.2}$ | $3.2 \pm_{0.7}$ |
| CLIP | $22 \pm_{3.5}$ | $26.8 \pm_{4.9}$ | $14.0 \pm_{6.8}$ | $14.8 \pm_{1.3}$ |
| LIV-EPIC | $29.3 \pm_{4.6}$ | $\mathbf{32.3} \pm_{5.8}$ | $15.1 \pm_{4.3}$ | $17.3 \pm_{6.6}$ |

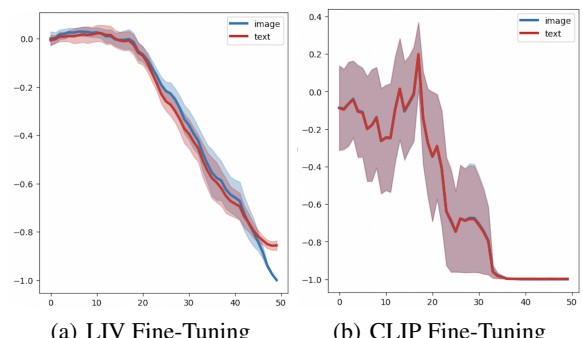

(a) LIV Fine-Tuning  (b) CLIP Fine-Tuning

*Figure 6.* **Qualitative Analysis:** LIV fine-tuning achieves both temporal coherence and semantic alignment; in contrast, CLIP fine-tuning over-aggressively aligns the goal frame-text pair that damages representations of earlier frames.

representations and train policies on top using LCBC.

**Baselines.** We consider using the **CLIP** InfoNCE objective (Eq. (4)) as well as the **VIP-I** objective (Eq. (1)) for fine-tuning. Note that these fine-tuning methods are ablations of LIV that focus only on semantic alignment and temporal-perception alignment, respectively. We also note that using CLIP to fine-tune CLIP is the current state-of-art (Goyal et al., 2022; Dong et al., 2022) on visual recognition tasks. We also consider direct fine-tuning using LCBC as a baseline. For an additional point of base model, we evaluate all fine-tuning methods on a representation learned via LIV on in-domain data from scratch ("Random").

**Results.** The results are shown in Table 1. We see that LIV fine-tuning is effective for all three model initializations, whereas the baseline ablations deliver mixed results. In particular, CLIP fine-tuning degrades performance in all cases except on the Random model in MetaWorld. This sub-par performance of CLIP fine-tuning, in stark contrast to concurrent works that have shown its effectiveness for image classification, reveals a fundamental difference be-

tween fine-tuning for *control* (e.g., robotic manipulation) and *recognition* (e.g., image classification). Robot demonstration data is typically scarce, and CLIP fine-tuning is wasteful since it ignores all but the last few frames of each demonstration. As such, CLIP fine-tuning is prone to overfitting and learning degenerate features that do map the goal image frame and the language command to the same embedding location but misrepresent other frames. This observation is supported by the relatively larger performance drop-off of CLIP fine-tuning on FrankaKitchen, as the robot dataset there is two orders of magnitude smaller than MetaWorld (12.5K vs. 1M).

This difference in dataset sizes also explains why VIP-I fine-tuning is reasonable on MetaWorld but very poor on FrankaKitchen, consistent with the findings in Ma et al. (2022b). As such, we have demonstrated that both terms in the LIV are indispensable for effective fine-tuning. LIV is uniquely effective at fine-tuning vision-language models under varying pretraining objectives, pretrained model qualities, and fine-tuning dataset sizes.

The final LIV fine-tuned models perform better when they started from better pre-trained models, so that the best combined system simply uses the LIV objective for *both phases*, pretraining as well as fine-tuning.

Finally, we find that fine-tuning directly using LCBC objective fails in all cases with policy losses exploding to infinity, despite our best efforts to stabilize training. Even warm-starting the policy network with frozen representations does not stabilize training. This is consistent with others' findings that shallow architectures are more suited for pure in-domain BC (Hansen et al., 2022); fine-tuning large vision-language models with BC is challenging.

**Qualitative Analysis.** We visually compare the fine-tuned LIV models via LIV and CLIP by overlaying the curves of the negative similarity score (i.e., $-\mathcal{S}(\phi(\cdot), \psi(\cdot))$) computed from all earlier frames to the goal frame and the goal text-command on demonstration fine-tuning data. We use one demonstration from each distinct task and average over all curves to produce Figure 6; individual task plots and the plots for the pre-trained LIV model are included in Appendix F. CLIP fine-tuning (Fig. 6(b)), as intended, maps the goal frame and the goal text to a near identical point in the representation space as the two curves almost perfectly overlap. However, the similarity scores of the intermediate frames exhibit highly unsmooth trend and variance, indicating that the representation does not preserve temporal coherence. In contrast, LIV (Fig. 6(a)) naturally preserves a structured representation, in which the visual and text similarity curves are near-linear, monotonic, and converge to similar locations, suggesting that the representation has successfully mapped the goal frame and text to similar em-

*Table 2.* **Planning with Learned Reward:** LIV-EPIC is both the strongest zero-shot and adapted reward model.

| Model | FrankaKitchen | MetaWorld |
|---|---|---|
| LIV-EPIC | $1.3 \pm 0.8$ | $29.7 \pm 4.7$ |
| LIV-EPIC (LIV Fine-Tuned) | $\mathbf{20.0} \pm 4.5$ | $\mathbf{55.2} \pm 5.5$ |
| CLIP | $0 \pm 0.0$ | $18.2 \pm 4.4$ |
| CLIP (LIV Fine-Tuned) | $15.2 \pm 4.6$ | $45.3 \pm 2.5$ |
| CLIP (CLIP Fine-Tuned) | $3.2 \pm 0.9$ | $30.7 \pm 3.3$ |
| LOREL | $9.6 \pm 3.0$ | $47.9 \pm 3.2$ |
| LOREL (R3M Initialized) | $16.8 \pm 3.8$ | $47.5 \pm 12.7$ |
| R3M | $8.8^* \pm 2.7$ | $18.3 \pm 7.7$ |
| R3M (R3M Fine-Tuned) | $16.1 \pm 4.2$ | $43.9 \pm 3.2$ |

beddings while preserving the temporal coherence of all earlier frames. This temporal consistency is crucial for effective representation as it automatically prevents incorrect observation aliasing and preserves feature scale across time for effective policy learning (Ma et al., 2022b; Nair et al., 2022b).

### 5.3. Reward Learning

While LIV fine-tuning has delivered strong results on LCBC, we recognize that a fundamental problem that in-domain fine-tuning is able to resolve is that of language grounding, which connects linguistic concepts to the visual attributes of the policy learning domain. This capability of distinguishing visual observations from the same domain based on an language input is precisely what is required for language-conditioned reward learning. Hence, we theorize that LIV's implicit value function that preserves both fine-grained semantic and temporal structure in its representation makes the ideal candidate for language-conditioned reward modeling.

**Baselines.** We compare to LOREL (Nair et al., 2022a), a state-of-art language-conditioned reward learning method that learns a classifier $f_\theta(o_0, o_t, l)$ for whether the progression from $o_0$ to $o_t$ completes task description $l$. In addition, we compare to R3M (Nair et al., 2022b), which incorporates a similar language-progression score function trained via contrastive learning. As the original LOREL does not leverage pre-trained visual representations, we also consider a variant of LOREL initialized with R3M model weights to improve its performance. Similarly, to circumvent the out-of-domain language grounding problem for pre-trained R3M, we consider a variant where we fine-tune the pre-trained R3M using the R3M objective on the same in-domain data used for LIV fine-tuning.

**Evaluations.** We evaluate all reward models in a model-based planning setup, in which a trajectory optimizer synthesizes a sequence of actions to be executed in the true environment based on scores from the utilized reward function. For all LIV models (pre-trained and fine-tuned), we use the potential-based reward as in Ma et al. (2022b):

$$R(o_t, o_{t+1}; l) := \mathcal{S}(\phi(o_{t+1}); \psi(l)) - \mathcal{S}(\phi(o_t); \psi(l)) \quad (6)$$

On the MetaWorld benchmark, we use the identical experimental setup as in Nair et al. (2022a), whereas on the FrankaKitchen benchmark, we closely follow the experimental protocol of Ma et al. (2022b). See Appendix E for more details on our model-based planning experiments. The aggregated success rate over all test instances are reported by benchmark in Table 2.

**Results.** As shown, LIV fine-tuning significantly improves the success rate over the base pre-trained LIV and CLIP models, and the fine-tuned LIV-EPIC achieves the best performance overall across both benchmarks. LOREL and R3M models both perform adequately with the respective modifications we have introduced, but they still trail behind LIV; likewise, CLIP's performance, even with LIV fine-tuning, is bottlenecked by the inferior performance of the pre-trained CLIP model.

These results illustrate the orthogonal, if not competing, nature of reward and representation capability of a vision-language model. While CLIP (CLIP Fine-Tuned) exhibits improved reward learning performance over pre-trained CLIP, in Section 5.2, we have shown that CLIP fine-tuning leads to far inferior representation backbone for policy learning. We believe this is because CLIP fine-tuning aligns the last frames with text goals, and the model-based planners we use evaluate action sequences based on only the reward of the last observation. In contrast, in imitation learning, the representation needs to be well-behaved for all intermediate observations, which CLIP fine-tuning impairs, as shown in Figure 6. LOREL is a reward learning algorithm, yet it is prone to overfitting when trained from scratch on small in-domain data (i.e., FrankaKitchen) and is most performant when initialized with a pre-trained representation. Finally, though R3M training involves learning a language-reward predictor, this predictor is trained only in service of the core visual representation training. We find that this predictor is inferior to even purely in-domain trained LOREL on Meta-World. LIV's implicit value learning paradigm gracefully combines both reward and representation learning in one unified objective and results in a flexible combined model that is highly effective across all evaluation settings. In Appendix E.2, we present additional analysis on these results, including performance comparison with scaled optimization budget as well as understanding R3M's zero-shot performance on FrankaKitchen.

## 6. Conclusion

We have presented the Language-Image Value Learning (LIV) algorithm. LIV is at once the first pre-training objective for control-oriented vision-language representations, a fine-tuning objective for domain-specific language grounding, and a language-conditioned task reward function. Trained on large generic human video datasets and fine-tuned on small simulated robotics datasets, LIV outperforms state-of-the-art approaches in each of three distinct evaluation settings. We will release pre-trained LIV models and fine-tuning code.

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

## A. Proof of Proposition 1

In this section, we provide a full proof of Proposition 1 in the main text. For ease of reading, we begin by reproducing the proposition.

**Proposition.** *Let the video distribution consist of solely degenerate videos of repeated frames that align with the text annotation, $D := \{v := ((g, g; l))\}$. Then, the VIP-L objective is equivalent to the InfoNCE objective up to a constant:*

$$\mathcal{L}_{\text{VIP-L}}(\phi, \psi) = \mathbb{E}_{p(g,l)} \left[ -\log \frac{e^{(1-\gamma)\mathcal{S}(\phi(g);\psi(l))}}{\mathbb{E}_{D(g')} \left[ e^{(1-\gamma)\mathcal{S}(\phi(g');\psi(l))} \right]} \right] + 1, \tag{7}$$

*where $p(g, l)$ is the distribution of goal frame and text pair.*

*Proof.* We begin with the VIP-L objective:

$$\mathbb{E}_{p(l)}[(1-\gamma)\mathbb{E}_{\mu_0(o;l)} \left[ -\mathcal{S}(\phi(o);\psi(l)) \right] + \log \mathbb{E}_{(o,o';l)\sim D} \left[ \exp\left( \mathcal{S}(\phi(o);\psi(l)) + 1 - \gamma\mathcal{S}(\phi(o');\psi(l)) \right) \right]] \tag{8}$$

We can massage this expression as follow:

$$\mathbb{E}_{p(l)}[\mathbb{E}_{\mu_0(o;l)} \left[ -(1-\gamma)\mathcal{S}(\phi(o);\psi(l)) \right] + \log \mathbb{E}_{(o,o';l)\sim D} \left[ \exp\left( 1 + (1-\gamma)\mathcal{S}(\phi(o);\psi(l)) \right) \right]], \tag{9}$$

assuming $o = o'$ in the log-sum-exp term.

Now, the joint distribution of language and initial-frame $p(l)\mu_0(o; l)$ reduces to the marginal distribution of goal-frame and text distribution $p(g, l)$ when the videos are just concatenations of the goal frames. Similarly, The language-conditioned distribution of successive intermediate frames $D(o, o'; l)$ reduces to the marginal distribution of goal frames $D(g')$ in the dataset. Plugging these substitution back into Equation (9) gives

$$
\begin{aligned}
&\mathbb{E}_{p(g,l)} \left[ -\log \frac{e^{(1-\gamma)\mathcal{S}(\phi(g);\psi(l))}}{\mathbb{E}_{D(g')} \left[ \exp\left( 1 + (1-\gamma)\mathcal{S}(\phi(g');\psi(l)) \right) \right]} \right] \\
=&\mathbb{E}_{p(g,l)} \left[ -\log \frac{e^{(1-\gamma)\mathcal{S}(\phi(g);\psi(l))}}{\mathbb{E}_{D(g')} \left[ e \cdot \exp\left( (1-\gamma)\mathcal{S}(\phi(g');\psi(l)) \right) \right]} \right] \\
=&\mathbb{E}_{p(g,l)} \left[ -\log \frac{e^{(1-\gamma)\mathcal{S}(\phi(g);\psi(l))}}{\mathbb{E}_{D(g')} \left[ \exp\left( (1-\gamma)\mathcal{S}(\phi(g');\psi(l)) \right) \right]} \right] + 1 \\
=&\mathbb{E}_{p(g,l)} \left[ -\log \frac{e^{(1-\gamma)\mathcal{S}(\phi(g);\psi(l))}}{\mathbb{E}_{D(g')} \left[ e^{(1-\gamma)\mathcal{S}(\phi(g');\psi(l))} \right]} \right] + 1
\end{aligned}
\tag{10}
$$

$\square$

## B. LIV Model Details

We implement LIV using the open-sourced CLIP architecture[1] without modifications; we use the modified ResNet50 (He et al., 2016) from CLIP for the vision encoder, and the CLIP Transformer (Vaswani et al., 2017; Radford et al., 2019) architecture for the language encoder. The training hyperparameters used during the pre-training and fine-tuning stages are listed in Table 3. During pre-training, we also incorporate the VIP-L objective, which we find to produce better pre-trained LIV models; we hypothesize that adding the explicit language-based VIP loss is instrumental in shaping the representation with semantic structure early on. During the fine-tuning stage, the same set of fine-tuning hyperparameters is used for fine-tuning CLIP as well as the ablation fine-tuning methods presented in Section 5.2.

Since LIV uses $-1$ as the constant fixed reward for all observations, the range of valid state value is $[\frac{-1}{1-\gamma}, 0]$; however, cosine similarity, as used in CLIP, has range of $[-1, 1]$. Thus, to be able to represent all possible values, we set $\mathcal{S}(\phi(\cdot), \psi(\cdot)) :=$

---

[1]https://github.com/openai/CLIP

$\frac{1}{1-\gamma}$CosineSimilarity($\phi(\cdot), \psi(\cdot)$). Coincidentally, with this choice of $\mathcal{S}$, the InfoNCE objective in LIV reduces to precisely the InfoNCE objective used in CLIP.

We pre-train LIV on EpicKitchen (Damen et al., 2018). We use the `EPIC-KITCHENS-100` version of the data and only utilize the RGB frames and text annotations from the dataset; the default frame rate in the raw dataset is used. The pre-training takes place on a node of 8 NVIDIA V100 GPUs.

*Table 3.* VIP Architecture & Pre-Training Hyperparameters.

|  | Pre-Training | Fine-Tuning |
|---|---|---|
| Model Initialization | CLIP | {LIV-EPIC, CLIP, Random} |
| Optimizer | Adam (Kingma & Ba, 2014) | Adam |
| Gradient Steps | 200000 | 10000 |
| Batch Size | 512 | 64 |
| Learning Rate | 0.00001 | 0.00001 |
| Weight Recay | 0.001 | 0.001 |
| Discount Factor $\gamma$ | 0.98 | {0.98, 0.96} |
| VIP-L objective | Yes | No |

## C. Environment Details

**MetaWorld.** The MetaWorld environment consists of a tabletop scene with a Sawyer robot that can interact with 4 objects, including a drawer, faucet, and two mugs distinguished by color. The dataset is collected by running a random policy for 50000 episodes with episode length 20; each episode is labeled with procedurally generated language descriptions that it achieves via computing pre-defined success criterion for each language-specified task. A single episode can solve many distinct tasks. In that case, the labeled description will be a concatenation of all atomic instructions that the episode has solved. The whole dataset contains 2311 unique descriptions, and the evaluation tests on 6 atomic instructions: `close drawer`, `open drawer`, `turn faucet right`, `turn faucet left`, `move black mug right`, `move the white mug left`.

**FrankaKitchen** The FrankaKitchen environment consists of a kitchen scene with a Franka robot that can interact with a variety of common household kitchen objects. We use the same 5-task split that was evaluated in Nair et al. (2022b) for visual imitation learning; the tasks as well as their language commands are listed in Table 4. For each task, we include 50 demonstrations, so the total size of the dataset is 250 episodes, where each episode is 50 environment steps long.

*Table 4.* FrankaKitchen Task Mapping

| Environment ID | Language Task |
|---|---|
| `kitchen_micro_open-v3` | `open microwave` |
| `kitchen_sdoor_open-v3` | `slide cabinet` |
| `kitchen_ldoor_open-v3` | `open left door` |
| `kitchen_knob1_on-v3` | `turn on stove` |
| `kitchen_light_on-v3` | `switch on light` |

## D. Language-Conditioned Imitation Learning

We present the LCBC imitation learning hyperparameters in Table 5. Because the dataset size in MetaWorld is significantly larger, we use a larger MLP architecture with bigger batch size. For each distinct evaluation task, we rollout for 50 episodes and record the success rate.

### D.1. Full Numeric Results

In Table 6, we present the full numeric results of our LCBC with pre-trained representations experiment.

*Table 5.* LCBC Hyperparameters.

|  | MetaWorld | FrankaKitchen |
|---|---|---|
| MLP Architecture | [1024, 1024, 1024] | [256, 256] |
| Non-Linear Activation | ReLU | ReLU |
| Optimizer | Adam | Adam |
| Gradient Steps | 200000 | 200000 |
| Batch Size | 4096 | 32 |
| Learning Rate | 0.001 | 0.001 |
| Proprioception | No | Yes |

*Table 6.* **Pre-Trained Representations for Language-Conditioned Imitation Learning:** LIV-EPIC achieves the highest average success rates across two distinct benchmarks and makes most effective use of its language embedding.

| model | FrankaKitchen | MetaWorld |
|---|---|---|
| **LIV-EPIC** | **29.3** $\pm$ 4.6 | **30.6** $\pm$ 5.0 |
| LIV-EPIC (One-Hot) | 17.6 $\pm$ 5.0 | 26.1 $\pm$ 5.5 |
| CLIP | 22 $\pm$ 3.5 | 19.4 $\pm$ 1.3 |
| CLIP (One-Hot) | 14.8 $\pm$ 0.7 | 28.6 $\pm$ 1.3 |
| VIP (BERT) | 18.0 $\pm$ 6.9 | 24.2 $\pm$ 3.0 |
| VIP (One-Hot) | 15.6 $\pm$ 6.2 | 28.3 $\pm$ 0.8 |
| R3M (BERT) | 18.7 $\pm$ 11.0 | 12.7 $\pm$ 3.9 |
| R3M (One-Hot) | 11.5 $\pm$ 1.9 | 18.1 $\pm$ 5.5 |

## E. Reward Learning

We describe our model-based planning experimental details. On MetaWorld, we use a cross-entropy Method (CEM) (Rubinstein & Kroese, 2004) planner to propose action sequences and employ the open-sourced SV2P (Babaeizadeh et al., 2017) visual dynamics model trained on the demonstration data to rollout the action sequences for optimization. On FrankaKitchen, as in Ma et al. (2022b), we use the ground-truth environment dynamics to for action rollouts and employ a model-path predictive integral (MPPI) (Williams et al., 2017) planner. On FrankaKitchen, due to the exploration challenge, we also warmstart the action search with a fixed open-loop sequence that brings the robot end-effector to the vicinity of the task object but does not perform the full commanded task.

### E.1. Hyperparameters

On MetaWorld, we use the open-sourced implementation of Cross-Entropy Method (CEM) on this environment released by (Nair et al., 2022a). On FrankaKitchen, we follow the practice of Ma et al. (2022b) and use a publicly available implementation of MPPI[2] with the default hyperparameters.

*Table 7.* Model-Based Planning Hyperparameters.

|  | MetaWorld | FrankaKitchen |
|---|---|---|
| Planner | CEM | MPPI (Williams et al., 2017) |
| Planning Horizon | 20 | 50 |
| # Proposed Action Sequences | 200 | 128 |
| Optimization Iteration | 1 | 1 |
| Dynamics Model | SV2P trained on in-domain dataset | Ground truth simulation |

---

[2]https://github.com/aravindr93/trajopt/blob/master/trajopt/algos/mppi.py

Table 8. LIV models consistently improve with increased planning budget; in contrast, baselines report mixed results.

| Model | MetaWorld (CEM Iterations=1) | MetaWorld (CEM Iterations=3) |
|---|---|---|
| LIV-EPIC | 29.7 | 34 |
| LIV-EPIC (LIV Fine-Tuned) | **55.2** | **57.8** |
| CLIP | 18.2 | 14.7 |
| CLIP (LIV Fine-Tuned) | 45.3 | 44.4 |
| CLIP (CLIP Fine-Tuned) | 30.7 | 34.4 |
| LOREL | 47.9 | 55.4 |
| LOREL (R3M Initialized) | 47.5 | 50.6 |
| R3M | 18.3 | 18.1 |
| R3M (R3M Fine-Tuned) | 43.9 | 50.8 |

Table 9. Performance Comparison Between Correct and Random Language Goals.

| Model | Correct Goal | Random Goal |
|---|---|---|
| LIV-EPIC | 1.3 | 1.0 |
| LIV-EPIC (LIV Fine-Tuned) | **20.0** | 0.0 |
| LOREL | 9.6 | 0.0 |
| LOREL (R3M Initialized) | 16.8 | 0.0 |
| R3M | 8.8 | 12.1 |
| R3M (R3M Fine-Tuned) | 16.1 | 0.0 |

### E.2. Additional Results & Analysis

**How does increasing planning budget affect model performance?** To further assess the capability of the various learned reward models, we repeat the model-based planning experiment on MetaWorld by increasing the CEM optimization iteration from 1 to 3. The results are shown in Table 8. We see that almost all models that are trained or fine-tuned on the in-domain data see performance increase with the fine-tuned LIV-EPIC standing as the best model. However, the pre-trained models (LIV-EPIC, CLIP, R3M), with the exception of LIV-EPIC, see performance degradation, suggesting that their reward models are in fact exploited by the stronger optimizer. Finally, we observe that LIV with 1 CEM iteration already performs as well as LOREL with 3 CEM iterations, suggesting that LOREL is more prone to "false nagatives", i.e. assigning low scores to good trajectories. These results highlight both LIV's ability for zero-shot and fine-tuning reward model.

**Why does R3M work well zero-shot on FrankaKitchen?** Interestingly, we find R3M to perform well zero-shot on FrankaKitchen (Table 2, achieving ≈ 9% success rate without any in-domain fine-tuning. Upon investigating this outcome however, we find that this result is an artifact of the specific way in which R3M was trained. In particular, R3M's pre-trained reward predictor has a bias for actions that induce visual change in the environment because it was pre-trained to output higher scores for frames that are farther apart in time, which typically correlate with larger visual changes in the scene. To confirm this, we repeat the same experiment on FrankaKitchen but this time with *random* language goals. The results are shown in Table 9. We see that R3M's performance remains surprisingly high, indicating that it does not depend at all on the language-based task specification. In contrast, other models' performance catastrophically decline. This indicates that R3M's language grounding is limited and often confuses completion of specific tasks with any indiscriminate visual changes in the environment. This finding is further supported by R3M's poor performance on the MetaWorld environment, in which random actions are enough to move the objects and induce large visual changes, and task completion requires more directed action, driven by more sophisticated language understanding. LIV-EPIC significantly outperforms R3M on MetaWorld and is the best zero-shot reward model overall on this benchmark.

## F. Representation Qualitative Results

In this section, we provide additional qualitative results on our pre-trained and fine-tuned models.

### F.1. EpicKitchen (Real)

We first visualize pre-trained LIV-EPIC on representative seen and unseen EpicKitchen videos by plotting the embedding curves with respect to the image (final frame of the video) and the text goal. In both seen and unseen splits, the three videos have annotations `open cabinet`, `open door`, and `open microwave`, respectively. The results are in Figure 7 and 8. For comparison purpose, we include the results for the CLIP model in Figure 9 and 10.

### F.2. FrankaKitchen (Sim)

In Figure 11, 12, 13, we present the embedding curves for LIV-EPIC, LIV-EPIC (LIV finetuned), LIV-EPIC (CLIP finetuned) on the FrankaKitchen tasks. As shown, LIV-EPIC, wihtout any in-domain fine-tuning, is able to competently capture visual progress but lacks language grounding to capture language goal progress. LIV fine-tuning is able to enable language-conditioned progress while improving visual temporal alignment. CLIP fine-tuning over-aggressively aligns the representations of the last frame and the text goal and collapses intermediate representations.

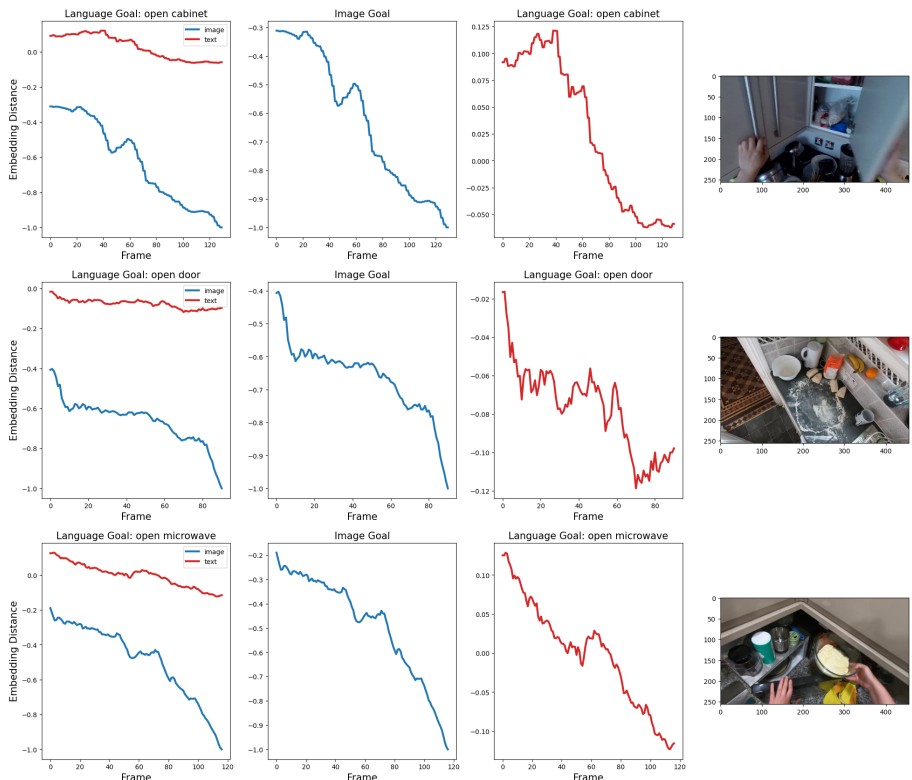

*Figure 7.* Pre-trained LIV-EPIC image and language goal reward curves on (seen) EpicKitchen videos.

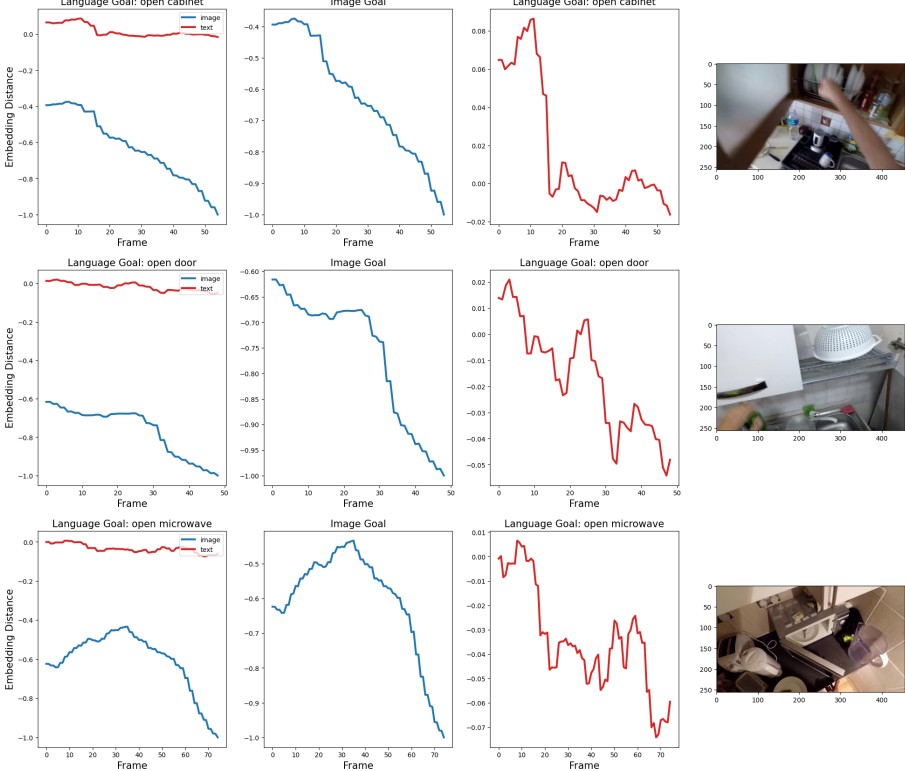

*Figure 8.* Pre-trained LIV-EPIC image and language goal reward curves on (unseen) EpicKitchen videos.

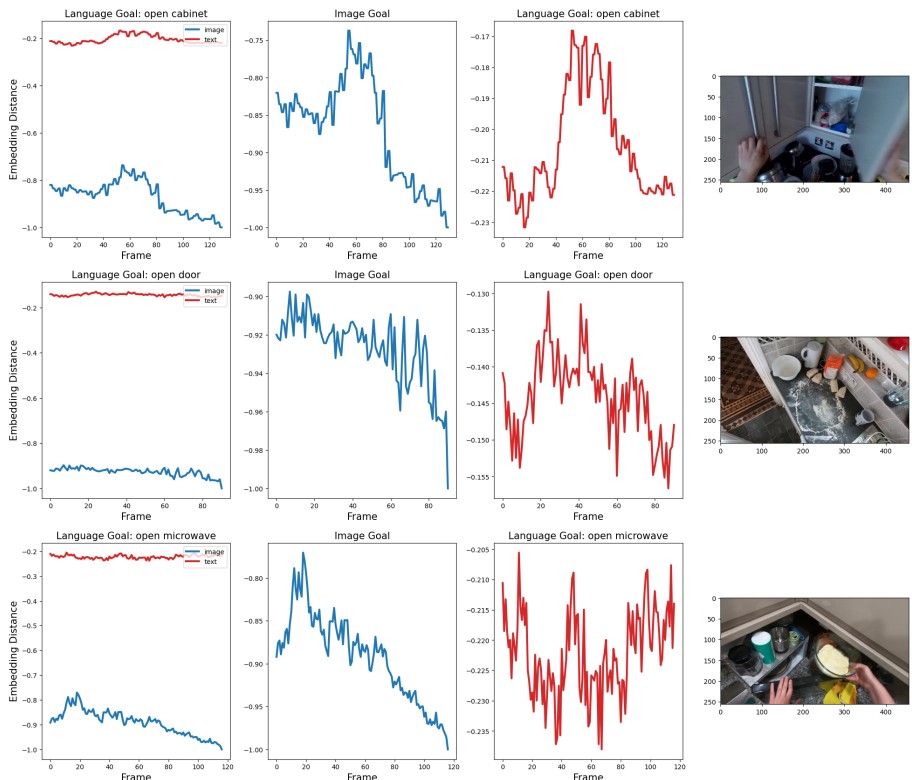

Figure 9. CLIP image and language goal reward curves on (seen) EpicKitchen (videos).

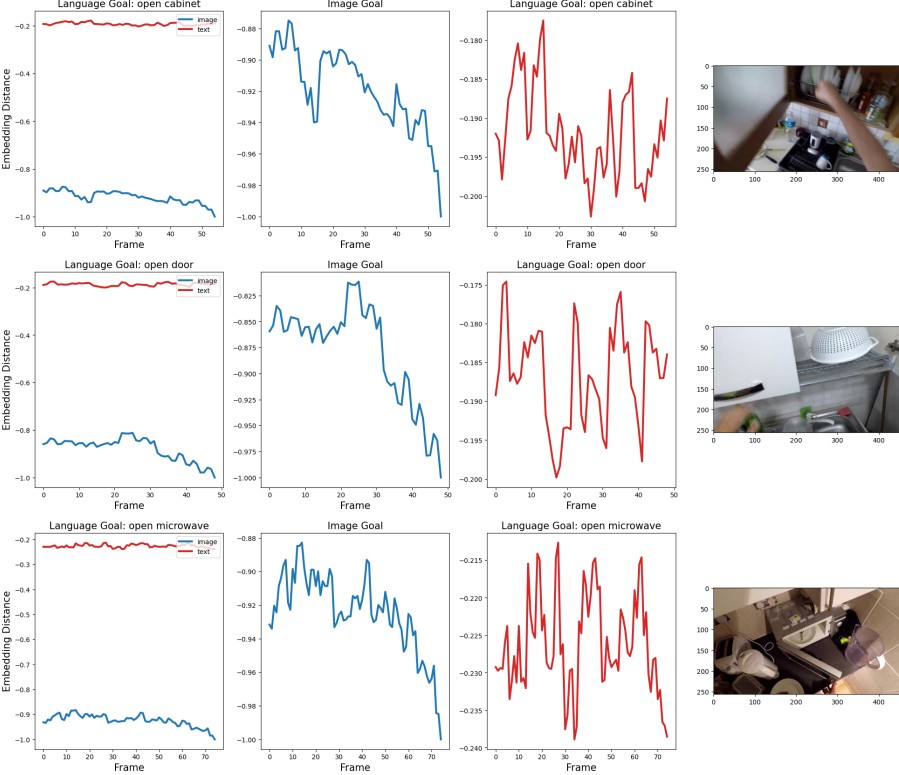

Figure 10. CLIP image and language goal reward curves on (unseen) EpicKitchen videos.

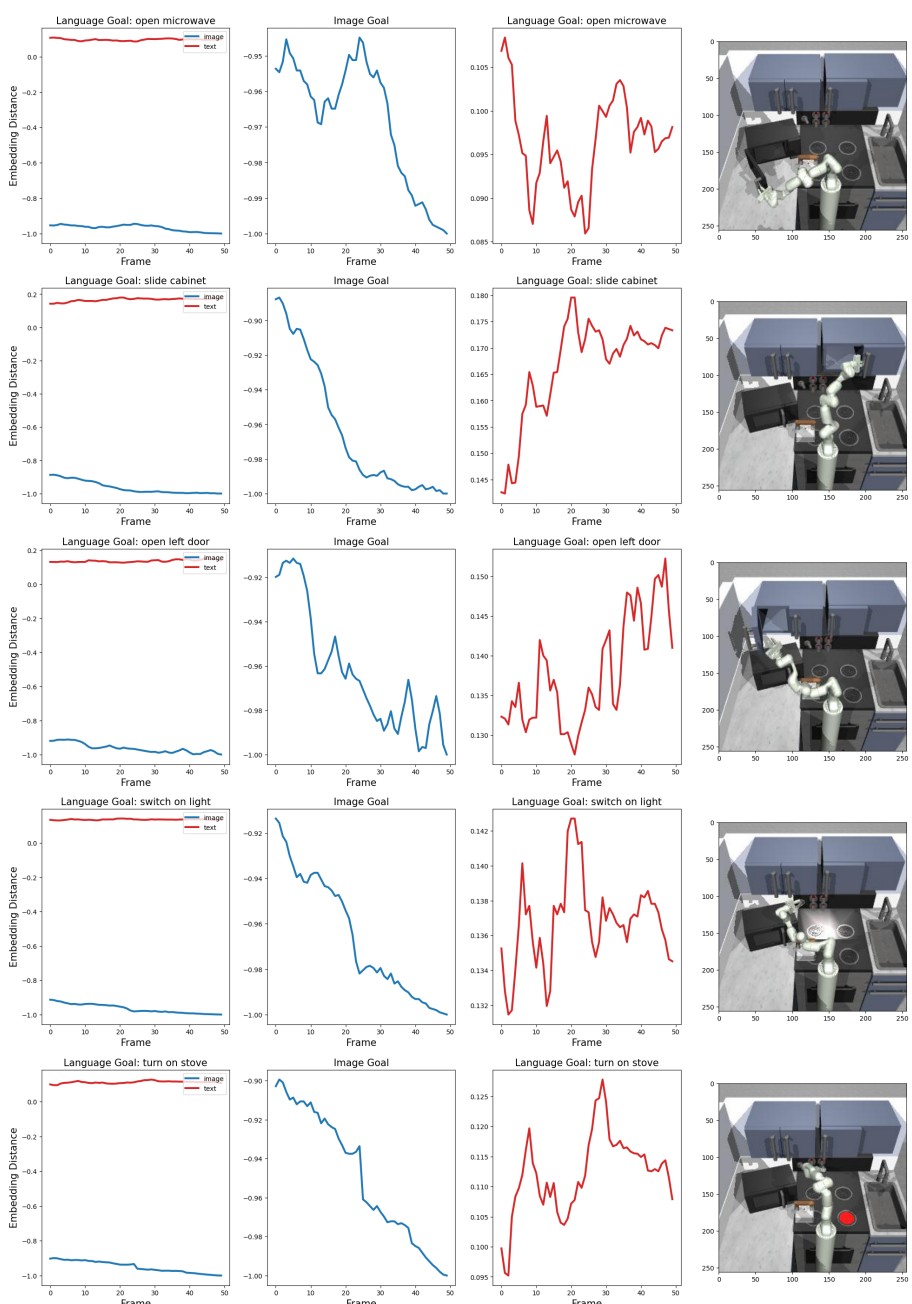

*Figure 11.* Pre-trained LIV-EPIC image and language goal reward curves on simulated FrankaKitchen tasks.

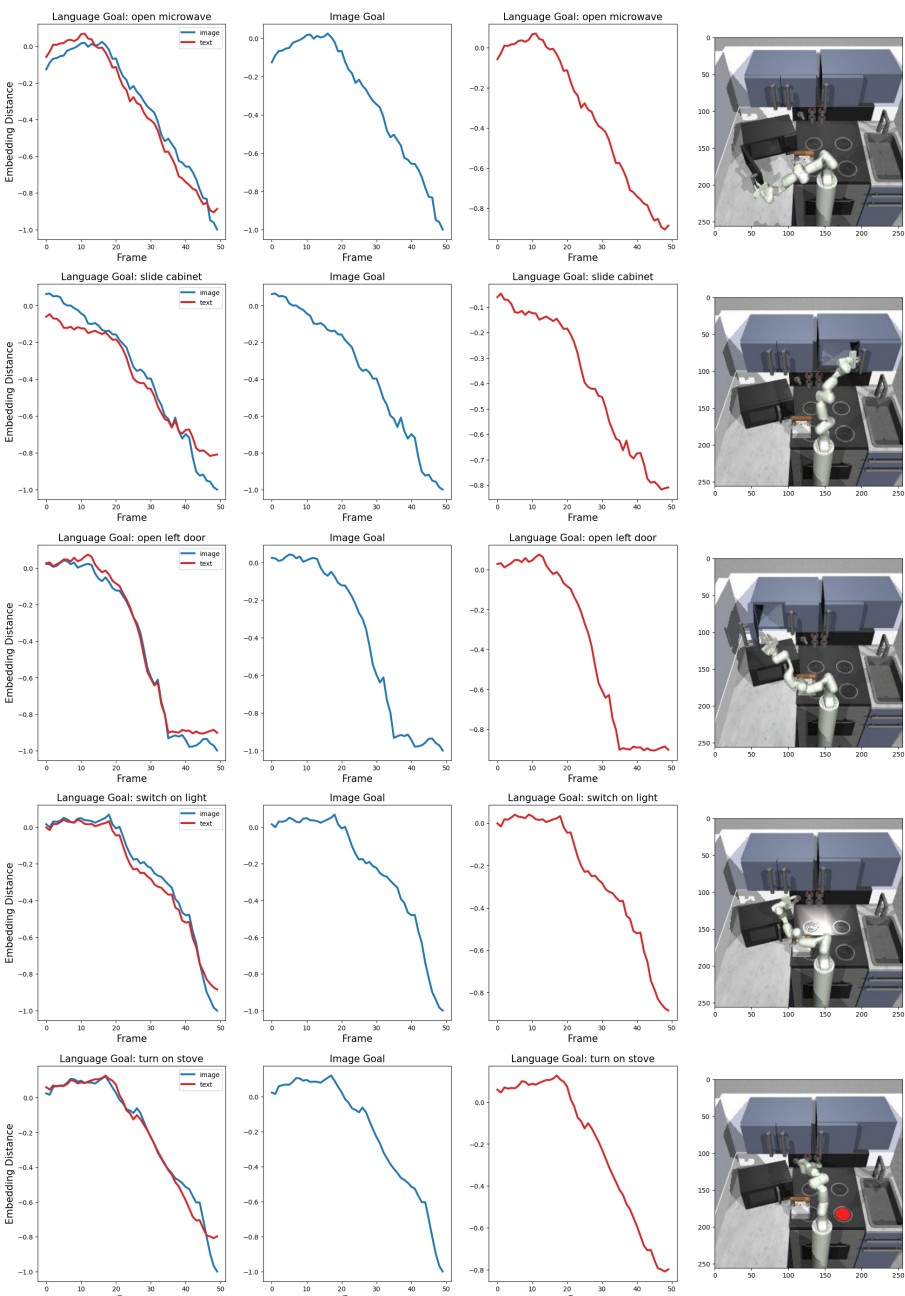

*Figure 12.* LIV-EPIC (LIV fine-tuned) image and language goal reward curves on simulated FrankaKitchen tasks.

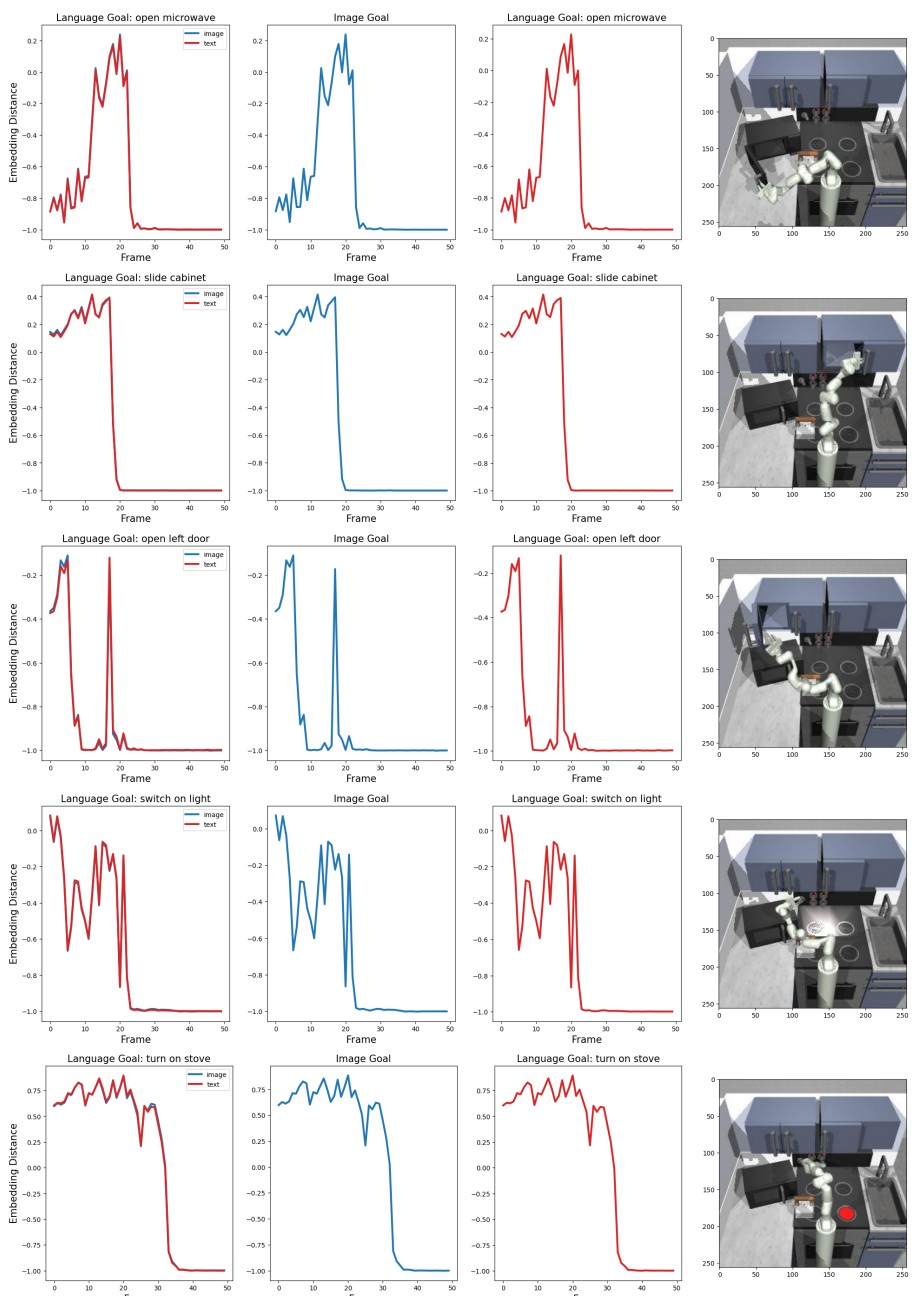

*Figure 13.* LIV-EPIC (CLIP fine-tuned) image and language goal reward curves on simulated FrankaKitchen.