# OpenReview forum: "LIV: Language-Image Representations and Rewards for Robotic Control"
_ICLR.cc/2023/Workshop/RRL — RRL 2023 Poster_

### Official Review · Reviewer_p8xm · 2023-02-27
**The authors propose LIV, a method for joint vision-language representation learning as well as reward learning for visual control that fits well to the topic of the workshop.**

**Rating:** 4
**Confidence:** 4

**Review:**

## Summary:

In this paper, the authors propose Language-Image Value Learning (LIV), an approach for joint vision-language representation learning as well as reward learning from action-free videos with text annotations. Their method combines the goal-conditioned image representation learning objective from Value Implicit Pre-Training (VIP) with the InfoNCE objective used in CLIP for vision-language representation learning. First, the authors pretrain their representation using LIV on the EpicKitchen dataset (a large dataset of narrated videos of humans completing kitchen tasks). Then, they use the frozen representation as a backbone for a policy trained on two continuous control benchmarks, FrankaKitchen and Meta-World. LIV can be used both as a pre-training and fine-tuning objective. The authors show that LIV pre-trained representations outperform related methods on the considered benchmarks.

## Strengths:
- The authors propose a new method LIV that builds on two recent approaches, VIP and CLIP.
- The authors stipulate clear criteria that pre-trained representations should fulfil and design their method accordingly:
    - Aligning the vision and language to permit grounded language specifications
    - Capture task-specific progress grounded in language
    - Domain generic pre-training + domain-specific fine-tuning
- The authors stipulate 3 research questions that they seek to answer in their experiments (pre-training, fine-tuning, language-conditioned reward specification). The subsequent experiments are well designed to answer their research questions.
- The presented results show that their method outperforms related methods compared in this work (significantly?)
- The LIV objective can be used both for pre-training, fine-tuning, as well as for reward learning.

## Weaknesses:
- Evaluation methodology:
    - The method is evaluated on a narrow set of down-stream tasks from Meta-World (6 tasks) and FrankaKitchen (5 tasks). Overall, the Meta-World benchmark contains 50 and FrankaKitchen 10 tasks. Also, the selected tasks are relatively simple, compared to the remaining tasks in the benchmark. Why are these particular ones selected? Showing results on more tasks would strengthen the paper.
    - The presented results exhibit high standard deviations (across 3 seeds) and significance tests are not conducted. Therefore, it is not clear whether LIV outperforms other approaches across settings. The word 'significantly' is used, however.
        - Table 1, Meta-World: LIV-EPIC vs. CLIP one-hot
        - Table 2, Meta-World: CLIP+LIV vs. Random+CLIP vs. Random+VIP-I vs. LIV-EPIC+LIV
        - Table 3, Meta-World: LIV-EPIC (Fine-Tuned) vs. LOREL
    - Reporting IQM and 95% confidence intervals (as proposed by https://arxiv.org/abs/2108.13264) would make the results more convincing, especially given the low number of seeds.
    - Adding additional baselines would be beneficial. Currently, the policy is trained via behaviour cloning. How would the BC baselines without pre-trained representations compare? How do established offline RL algorithms compare?
    - The authors state (lines 305-311), that they evaluate two trained policy checkpoints at the midpoints of training and end of training, and select the higher scores. What's the reasoning behind this?
- Without task-information, LIV performs considerably better than competitor methods. The authors make the argument that the learned language representation is better suited for identifying the tasks. When provided with additional tasks information (One-Hot), the competitor methods all improve. However, for LIV, the opposite is the case. Why is that? Shouldn't this result in even better performance?
- The authors make the argument that clip fine-tuning performs is wasteful, as it ignores all but the last few frames of each demonstration. This results in poor performance when a small number of samples is available (FrankaKitchen). Why wouldn't it be possible to use more frames across the demonstration? Then the performance gap may narrow.
- Lacking comparison of different Vision and language backbones.
- For the reward learning experiments, why is this specific model-based planning setup selected?
- Minor:
    - Figure 3: Add better descriptions, i.e., performance improvement over what? (Performance when task info is provided)
    - Figure 4: what does 'over its ablations' mean?
    - Figure 5: Axis descriptions. Would some sort of cluster analysis (e.g., via t-SNE) make this point stronger?
    - Table 3: make additional column for the fine-tuning approach. Currently, it's written in parentheses and it's a bit confusing.
    - Description of One-Hot setting: in Section 5.1 it is unclear what One-Hot refers to. Add a short description of what this means (i.e., that the task ID is provided?).
    - Use same font-size in Figures.

## Overall review:
In this paper, the authors propose LIV a method for joint vision-language representation learning as well as reward learning for visual control. Currently, the empirical results look promising, but are still limited in scope. Overall, we believe the papers fits the workshop well.

---

### Official Review · Reviewer_VvBC · 2023-02-28
**Good paper**

**Rating:** 3
**Confidence:** 5

**Review:**

Summary:
This paper aims to learn the first control-centric vision-language representation for general-purpose control. To achieve it, the author proposed the LIV, a unified objective for vision-language representation and reward learning from action-free videos with text annotations. The experimental results in some target domain environments validate the effectiveness of the proposed approach, especially in the comparison of CLIP.

Overall, I really liked this paper. And I think it is very important to learn a general control representation in RL, and this paper attempts to achieve it.

Strong points:
1. The motivation of this paper is exciting.
2. The proposed LIV approach to learning the control-centric vision-language representation is making sense.
3. The experimental results validate the effectiveness of the proposed approach.

Weak points:
1. I can not find the details of reward learning.